## [Peer Review File · The EMBO Journal]

Poxvirus dsDNA genomes differentially activate AIM2 or NLRP3 inflammasomes in human primary cells

Yonas Tesfamariam, Maria Christensen, Stefan Diehl, Tabea Klein, Julius Lingnau, Sabine Normann, Elena Hagelauer, Miriam Herbert, Sophie Reimer, Richa Joshi, Pujan Engels, Steffen Pritzl, Pietro Fontana, Thomas Zillinger, Gunther Hartmann, Anna-Maria Eis-Hübinger, Martin Lam, Klaus Walgenbach, Felix Meissner, Hao Wu, and Florian Schmidt

Corresponding author(s): Florian Schmidt (fschmidt@uni-bonn.de)

Review Timeline:

Submission Date:	8th Nov 24
Editorial Decision:	8th Jan 25
Revision Received:	17th Aug 25
Editorial Decision:	17th Oct 25
Revision Received:	27th Oct 25
Accepted:	24th Nov 25

Editor: Ioannis Papaioannou

Transaction Report:

Dear Dr. Schmidt,

Thank you for submitting your manuscript EMBOJ-2024-119572 for consideration by The EMBO Journal, and for your patience during peer review. Your manuscript has now been seen by three experts in the field, and we have received the full set of their comments, which are included below.

As you will see, the referees recognize that the work is interesting and well-developed, the results significant and novel, and the manuscript well-prepared. While the referees raise no major concerns, they provide a number of constructive suggestions for further clarification and strengthening of the work by adding a few more experiments and improving the presentation of some data. We agree with the referees that the manuscript and its impact on the field would benefit from the suggested improvements.

Given the referees' positive comments and recommendations, I would like to invite you to submit a revised version of the manuscript along with a detailed point-by-point response addressing all referees' comments. I should add that it is The EMBO Journal policy to allow only a single round of major revision, and acceptance of your manuscript will therefore depend on the completeness of your responses in this revised version. Please let me know if you have any questions or comments that you would like to discuss with me.

We generally allow three months as standard revision time (April 7, 2025). As a matter of policy, competing manuscripts published during this period will not negatively impact our assessment of the conceptual advance presented by your study. However, we request that you contact us as soon as possible upon publication of any related work, to discuss how to proceed. Should you foresee a problem in meeting this three-month deadline, please let us know in advance and we may be able to grant an extension.

Thank you for the opportunity to consider your work for publication in The EMBO Journal. I look forward to your revision.

Best regards,

Ioannis

Instructions for preparing your revised manuscript

1. When you are ready to submit the revision, please upload:

- A Word file of the manuscript text (including legends of main Figures, EV Figures and Tables). Please make sure that changes are highlighted (or "tracked") to be clearly visible.

- Individual production-quality figure files (one file per figure). When assembling your figures, please refer to our figure preparation guidelines in order to ensure proper formatting and readability in print as well as on screen:

If the data shown in a figure are obtained from n {less than or equal to} 2, please use scatter plots showing the individual data points.

- i. the name of the statistical test used to generate error bars and P values
- ii. the number (n) of independent experiments (please specify technical or biological replicates) underlying each data point (discussion of statistical methodology can be reported in the Materials and Methods section, but figure legends should contain a basic description of n , P , and the test applied)
- iii. the nature of the bars and error bars (s.d., s.e.m.).

- A point-by-point response to the referees' comments, with a detailed description of the changes made (as a word file). All referees' concerns must be fully addressed and their suggestions taken on board. When preparing your letter of response to the referees' comments, please bear in mind that this will form part of the Review Process File and will therefore be available online

to the community. Please note that you have the possibility to opt out of the transparent process at any stage prior to publication by letting the editorial office know (contact@embojournal.org); if you do opt out, the Review Process File link will point to the following statement: "No Review Process File is available with this article, as the authors have chosen not to make the review process public in this case.". For more details on our Transparent Editorial Process, please visit our website: <https://www.embopress.org/page/journal/14602075/authorguide#transparentprocess>

- Expanded View (EV) files (replacing Supplementary Information) that are collapsible/expandable online. A maximum of 5 EV Figures can be typeset. EV Figures should be cited as "Figure EV1, Figure EV2" etc. in the text, and their respective legends should be included in the manuscript file after the legends of regular figures. See detailed instructions regarding Expanded View files here:

- For the figures that you do NOT wish to display as Expanded View figures, they should be bundled together with their legends in a single PDF file called "Appendix", which should start with a short Table of Contents (including page numbers). Appendix figures should be referred to in the main text as: "Appendix Figure S1, Appendix Figure S2" etc. Please see detailed instructions here: <https://www.embopress.org/page/journal/14602075/authorguide#expandedview>

- A complete author checklist, which you can download from our author guidelines (<https://www.embopress.org/page/journal/14602075/authorguide>). Please note that the checklist will also be part of the Review Process File.

2. Please note that no statistics should be calculated and shown in Figures if $n=2$. Please also note that each p value should be reported as an exact value.

3. Before submitting your revision, primary datasets (and computer code, where appropriate) produced in this study need to be deposited in appropriate public databases (see <https://www.embopress.org/page/journal/14602075/authorguide#dataavailability>).

In particular, we kindly request you to deposit the mass spectrometry data produced in your study to an appropriate database. The accession numbers, database, and the specific URLs (links) should be listed in a formal "Data availability" section (placed after Methods), following the example below:

"The RNA-seq datasets produced in this study are available in the following database:
Gene Expression Omnibus GSE46843 (<https://www.ncbi.nlm.nih.gov/geo/query/acc.cgi?acc=GSE46843>)"

*** All links should resolve to a page where the data can be accessed. ***

*** Please remember to provide in the Data availability section of your revised manuscript reviewer passwords if the datasets are not yet public. ***

*** The Data Availability Section is restricted to new primary data that are part of this study. In case you have no data that require deposition in a public database, please state so instead of referring to the database: "Our study includes no data deposited in public repositories." under the heading "Data availability". ***

4. Please check that the title and the abstract of the manuscript are brief, yet explicit, even to non-specialists. The length of the title should not exceed 100 characters, and the abstract should be a single paragraph not exceeding 175 words.

5. Please also note our reference format: <https://www.embopress.org/page/journal/14602075/authorguide#referencesformat>.

7. Please remember: digital image enhancement is acceptable practice, as long as it accurately represents the original data and conforms to community standards. If a figure has been subjected to significant electronic manipulation, this must be noted in the figure legend or in the "Materials and Methods" section. The editors reserve the right to request original versions of figures and the original images that were used to assemble the figure.

8. Our journal encourages inclusion of data citations in the reference list to directly cite datasets that were obtained from public databases. Data citations in the article text are distinct from normal bibliographical citations and should directly link to the database records from which the data can be accessed. In the main text, data citations are formatted as follows: "Data ref: Smith et al, 2001" or "Data ref: NCBI Sequence Read Archive PRJNA342805, 2017". In the Reference list, data citations must be labeled with "[DATASET]". A data reference must provide the database name, accession number/identifiers, and a resolvable link to the landing page from which the data can be accessed at the end of the reference. Further instructions are available at: <https://www.embopress.org/page/journal/14602075/authorguide#referencesformat>.

9. We request authors to consider both actual and perceived competing interests. Please review our policy (<https://www.embopress.org/page/journal/14602075/authorguide#conflictsofinterest>) and update your competing interests statement if necessary. Please name this section 'Disclosure and competing interests statement' and place it after the Acknowledgements section.

10. Please note that all corresponding authors are required to provide an ORCID ID upon submission of a revised manuscript (<https://orcid.org/>). Please find instructions on how to link your ORCID ID to your account in our manuscript tracking system in our Author guidelines (<https://www.embopress.org/page/journal/14602075/authorguide#authorshipguidelines>).

11. We use CRediT to specify the contributions of each author in the journal submission system. CRediT replaces the author contribution section, which should be removed from the manuscript. Please use the free text box to provide more detailed descriptions. See also guide to authors: <https://www.embopress.org/page/journal/14602075/authorguide#authorshipguidelines>.

13. We would also welcome the submission of cover suggestions or motifs to be used by our Graphics Illustrator in designing a cover.

14. Please use the link below to submit your revision:
<https://emboj.msubmit.net/cgi-bin/main.plex>

Referee #1:

Schmidt and colleagues clarify the role of AIM2 and NLRP3 in sensing dsDNA poxviruses VACV and MPXV in different primary human cell types. The approaches to do this are elegant and use cutting edge technology of employing inhibitory nanobodies to implicate AIM2. The novel aspects of the study are: (1) The generation, characterisation and use of AIM2 nanobodies (2) Clarification of the cell type specific role of AIM2 and NLRP3 in inflammasome sensing of VACV (3) Demonstration of inflammasomes involved in sensing monkey pox virus (4) a role for the poxvirus immune evasion protein poxin in inhibiting NLRP3 inflammasome activation (presumably by inhibiting upstream cGAS function). There are a few simple experiments that would help further clarify their model:

(1) Show in each cell type where IFN γ renders cells able to sense VACV via AIM2 that AIM2 protein expression is actually increased by IFN γ . Currently this is only shown for THP-1 cells. Given part of the novelty of the study is the clarifying work in primary human cells, AIM2 protein expression should be shown in these (MDMs, NHEKs)

(2) B2R encodes poxin which degrades cGAMP, but many viral immune evasion proteins have multiple functions, so the relationship between B2 effect and a role for cGAS may not always be direct. Given the importance in the paper of clarifying the role of AIM2 versus cGAS-STING-NLRP3 for inflammasome activation, it would be good to compare the B2R mutant virus to WT virus not only in the cell types where NLRP3 is implicated a central, but also in the cell types where AIM2 and not NLRP3 is implicated as central.

(3) Related to (3), there are many small molecule cGAS and STING inhibitors that can be easily used in the primary cells to further clarify a role for cGAS/STING in poxvirus-mediated inflammasome activation. It would be interesting to see if there is a direct correlation between the role of cGAS/STING and NLRP3 by using cGAS/STING small molecule inhibitors as well as CRID3.

Referee #2:

Reviewer Comments:

The manuscript titled "Incoming poxvirus dsDNA genomes differentially activate AIM2 or NLRP3 inflammasomes in human cells" by Tesfamariam et al. explores the molecular pathways leading to inflammasome activation in human primary cells infected with poxviruses. Using innovative tools such as antagonistic AIM2 nanobodies and engineered vaccinia viruses, the authors demonstrate cell-type-specific inflammasome activation: AIM2 inflammasomes are triggered in IFN- γ -treated macrophages and keratinocytes, while NLRP3 inflammasomes are activated in CD14+ monocytes. Importantly, the study establishes that incoming poxvirus DNA genomes directly activate AIM2 inflammasomes and are critical for NLRP3 activation in monocytes. These

findings provide valuable insights into the mechanisms of poxviral modulation of host immune responses.

Strengths:

The manuscript is well-structured, with clearly presented results supported by a comprehensive experimental approach. It employs diverse methods, including engineered vaccinia viruses, AIM2 nanobodies, and reconstituted cell systems, to dissect the interplay between poxvirus infection and inflammasome activation. The development and characterization of novel AIM2 nanobodies represent a significant contribution to addressing the lack of specific AIM2 inhibitors.

Tesfamariam et al. present a thorough and impactful study on poxvirus-induced inflammasome activation, with significant implications for understanding host-pathogen interactions. Addressing the outlined suggestions would further enhance the manuscript's clarity and scientific impact.

Figure 1. VACV infection induces inflammasome activation in IFN- γ treated macrophage-like cell line. (A-D) The authors used PMA-differentiated THP-1 macrophages, either pre-treated with LPS, IFN- γ , IFN- α , followed by infection with VACV WR WT. They measured IL-1 in the supernatant, LDH release, membrane permeabilization quantified by the uptake of DNA dye. The results showed that in IFN- γ pretreated macrophage-like cells, VACV WR WT infection induced inflammasome activation leading to IL-1 secretion and cell death via pyroptosis. (E) The authors generated a THP-1 reporter cell line expressing caspase-1CARD-EGFP (C1C-EGFP), which allowed quantification of inflammasome assembly upon LPS plus Nigericin treatment. I am not sure why the authors did not include an example of dot plot of flow analysis of IFN- γ -pretreated THP-1 reporter cells infected by VACV WR WT in Figure 1E. But they did include a quantification of % speckling cells in Fig. 1F. (G) showed engineering of VACV expressing C1C-EGFP or EGFP. They showed that VACV C1C-EGFP infection of THP-1 WT cells pre-treated with IFN- γ induced IL-1 β and LDH release. Is there any difference between VACV WR WT vs. VACV C1C-EGFP in inducing IL-1 β secretion, LDH release and uptake of DNA dye in IFN- γ -pretreated THP-1 cells?

Figure 2. Poxvirus infection induces inflammasome activation in primary human cell types. The authors investigated whether vaccinia infection in primary human cell types, including human monocytes-derived macrophages (differentiated in GM-CSF or M-CSF), normal human epidermal keratinocytes (NHEKs), CD14⁺ monocytes induce inflammasome activation in comparison to PMS-differentiated THP-1 macrophage-like cells, with or without IFN- γ pre-treatment. The labels for the condition did not appear until Fig. 2E. It would be helpful to include the labels in Fig. 2A or 2B. The authors did not provide a mechanistic explanation why GM-CSF and M-CSF hMDMs are different in their response to VACV C1C-EGFP in inducing inflammasome activation. In addition, unlike in Figure 1, the authors did not mention about IL-1 β secretion, LDH release and uptake of DNA dye in this experiment with human primary cells. They only focused on the formation of specks in the infected cells. In Fig. 2F and 2G, the authors compared the infection efficiency and inflammasome induction of MPXV and VACV in NHEK cells and THP-1 cells expressing C1C-EGFP. They found that MPXV infection is weaker in inducing inflammasome activation compared with VACV. It would be helpful to evaluate IL-1 β secretion, LDH release and uptake of DNA dye in this experiment to get a full picture of inflammasome activation. In Fig. 2H, the authors examined induction of specks by VACV C1C-EGFP and VACV EGFP in THP-1 cells with various deficiencies in innate immune sensors with or without IFN- γ pre-treatment. They found that in THP-1 cells pre-treated with IFN- γ , the induction of specks is dependent on AIM2 and ASC, but not on NLRP3. Again, it would be helpful to include IL-1 β secretion, LDH release and uptake of DNA dye in this experiment.

Figure 3. Identification and characterization of AIM2 nanobodies. This is the novel aspect of this study. The authors provided rationale for developing AIM2 nanobodies for this study, as there are no available AIM2-specific inhibitors. They immunized alpaca with MBP-AIM2PYD L10AL111A, a mutant of the PYD that cannot polymerize into filaments, and cloned sequences for the variable domains of heavy chain-only antibodies (VHHs) into a phagemid vector. They used phage display method to identify AIM2PYD-specific nanobodies. A total of 18 hits were identified. Out of the 18 candidates, three nanobodies were selected, VHHAIM2-1, VHHAIM2-2, VHHAIM2-3, based on their function in AIM2 binding, but not NLRP1CARD binding. Competitive ELISA was performed using VHH-LPETG, which revealed that VHHAIM2-1 and VHH AIM2-3 share an overlapping epitope. Does the author know the identity of the epitope recognized by both VHHAIM2-1 and VHH AIM2-3?

Figure 4. AIM2 nanobodies inhibit inflammasome assembly by preventing oligomerization of AIM2PYD filaments. The authors used a transfection system to evaluate whether cytosolic expression of AIM2PYD nanobodies interfered with AIM2 inflammasome assembly and AIM2PYD-EGFP filament formation. The results confirmed that AIM2 nanobodies, VHHAIM2-1, VHHAIM2-2, VHHAIM2-3, either monovalent or bivalent, are able to inhibit AIM2 inflammasome activation and AIM2PYD filament oligomerization. Would it be possible to use a biochemical method to show AIM2PYD filament formation and disruption in the presence of AIM2 nanobodies, in addition to the imaging?

Figure 5. VACV encoded antagonistic AIM2 nanobodies reveal cell-type-specific inflammasome activation in human primary cells. The authors engineered a vaccinia virus (VACV) strain that expresses C1C-EGFP under the J2R early promoter and bivalent nanobodies driven by a synthetic early/late promoter. Their findings demonstrate that VACV expressing a bivalent control VHH targeting NP1 failed to inhibit speck formation. In contrast, VACV expressing bivalent VHHs targeting ASC or AIM2-1 effectively blocked speck formation in GM-CSF-derived human monocyte-derived macrophages (GM-CSF-hMDMs) and normal human epidermal keratinocytes (NHEKs). However, in CD14⁺ monocytes, VACV expressing bivalent ASC inhibited speck formation, whereas VACV expressing bivalent AIM2-1 did not. Notably, speck formation in CD14⁺ monocytes was effectively inhibited by the NLRP3 inhibitor CRID3. These results suggest that VACV infection triggers distinct inflammasome

activation pathways depending on the cell type. In IFN- γ -treated GM-CSF-hMDMs and NHEKs, VACV induces AIM2-dependent inflammasome activation, whereas in IFN- γ -treated CD14⁺ monocytes, inflammasome activation occurs via NLRP3. It would be helpful to present representative images in addition to the quantification graphs.

Figure 6. VACV infection is directly sensed by reconstituted AIM2 inflammasomes.

The authors demonstrated that IFN- γ treatment induces AIM2 expression in various primary cells and PMA-differentiated THP-1 cells. While AIM2 RNA levels were shown for primary cells, both RNA and protein levels were presented for THP-1 cells. It would be helpful for the authors to also include protein level data for AIM2 in CD14⁺ monocytes, NHEKs, GM-CSF-macrophages, and M-CSF macrophages, with and without IFN- γ treatment, to provide a more comprehensive analysis.

To further investigate AIM2 function, the authors generated THP-1 AIM2 knockout (KO) cells and transduced them with doxycycline (dox)-inducible AIM2 constructs. They observed that VACV-C1C-EGFP infection in Dox-treated THP-1 cells induced speck formation irrespective of IFN- γ treatment. This finding suggests that overexpression of AIM2 can bypass the requirement for IFN- γ priming.

Additionally, the authors reconstituted Dox-inducible AIM2 and NLRP3 in HEK293 cells engineered to express ASC-EGFP but lacking cGAS and STING. Upon infection with VACV or MPXV, they found that both viruses induced speck formation in HEK-AIM2i cells, but not in HEK-NLRP3i cells, in the presence of Dox. These results indicate that viral DNA directly activates AIM2 inflammasomes in these reconstituted cells. It would be helpful to present representative images in addition to the quantification graphs.

Figure 7. VACV incoming genomes are sufficient for the activation of inflammasomes, independent of viral DNA replication.

Lastly, the authors evaluated whether incoming viral genomes are sufficient to activate inflammasomes. To test this, they used the translation inhibitor cycloheximide (CHX) to block all viral gene expression and the DNA replication inhibitor cytosine arabinoside (AraC) to inhibit viral DNA replication and intermediate and late gene expression. In THP-1 cells constitutively expressing C1C-TagBFP and pre-treated with IFN- γ , infection with VACV expressing EGFP under an early promoter induced inflammasome activation. This activation was blocked by CHX but not by AraC, indicating that incoming viral genomes are sufficient to activate the AIM2 inflammasome in differentiated THP-1 cells. Furthermore, confocal imaging revealed that released viral DNA co-localized with C1C-mCherry. It would be helpful to include quantification data from this experiment and also include imaging from CHX-treated samples.

The authors then investigated which viral structures trigger inflammasome assembly in CD14⁺ monocytes, given that activation occurs through the NLRP3 inflammasome in these cells. They observed that VACV Δ B2R C1C-EGFP induced a higher percentage of specks compared with VACV C1C-EGFP, and this effect was moderately reduced in the presence of the STING inhibitor H-151. AraC treatment did not affect speck induction by either VACV C1C-EGFP or VACV Δ B2R C1C-EGFP. To provide a comprehensive understanding of inflammasome activation by VACV C1C-EGFP and VACV Δ B2R C1C-EGFP in CD14⁺ monocytes, it would be important to include measurements of IL-1 β secretion, LDH release, and uptake of DNA dye in this experiment. Additionally, incorporating CHX treatment in CD14⁺ monocytes would help demonstrate the importance of viral gene expression in NLRP3 inflammasome activation in these cells.

Referee #3:

In this manuscript, the authors dissect inflammasome activation in various cell types as an immune response to Vaccinia virus and Mpox virus. Within immortalized and primary myeloid cell types, AIM2 inflammasomes are activated. However, in monocytes, VACV activates the NLRP3 inflammasome. The investigators develop novel AIM2 nanobodies, designed to block the polymerization and nucleation of inflammasomes - it stands as the first specific inhibitor of AIM2. The authors utilize novel techniques such as their Casp-1 CARD EGFP construct inside of the VACV genome to assess inflammasome activation - and was necessary because of additional immunomodulators within the VACV/Mpox genome can alter the readout of other assays such as pyroptosis and IL-1B release. The incorporation of the AIM2 nanobody sequence into the VACV genome was also an interesting experimental design - which worked to show the AIM2 dependence in some cell types. Lastly, the activation of inflammasomes was shown to be a result of initial infection, virus-replication independent - suggesting the AIM2 binds directly to the viral genome. The NLRP3 inflammasome activation is modulated by the activity of host STING, where VACV-encoded poxins-schlafen proteins interfere with STINGs cross-talk with NLRP3 inflammasomes and lysosomal rupture. All-in-all, this investigation yields critical insights into the immunomodulation of host cells by VACV, and provides a mechanistic explanation for inflammasome activation in various cell types.

- specific major concerns essential to be addressed to support the conclusions
No major concerns were noted.

- minor concerns that should be addressed

Three separate instances throughout the paper suggest that experiments were performed and data was collected but not shown. There is also a manuscript in process that is cited early on regarding the development of in-house nanobodies. I would suggest to compile this data into an additional supplementary figure.

Dear Dr. Papaioannou, dear reviewers,

We would like to thank you for the time and effort to review our manuscript and to make helpful suggestions to improve our study. We have now prepared a revised version of the manuscript with additional controls and data. We address the reviewer requests for additional readouts for inflammasome activation and cell death, show the induction of AIM2 by IFN- γ using immunoblot, and provide biochemical evidence for the inhibition of AIM2^{PYD} oligomerization. As detailed below, we found that the original stock of VACV Δ B2R (lacking the cGAMP esterase) was not a clean knockout and produced two independent, validated knockout strains with interesting phenotypes. Yet, as they do not seem to be helpful to address the role of STING and NLRP3 in VACV-induced inflammasome assembly in monocytes, we suggest removing this panel from the manuscript.

The overall message of the manuscript remains the same: We developed the (to our knowledge) first specific inhibitor for AIM2 as well as versatile engineered VACV strains encoding the inflammasome sensor C1C-EGFP. We used these new tools to reveal that incoming poxvirus genomes activate NLRP3 inflammasomes in monocytes and AIM2 inflammasomes in IFN- γ -treated monocyte-derived macrophages and keratinocytes. We think that the revisions improved the manuscript substantially and hope that you find the revised version suitable for publication in the EMBO Journal.

Referee #1:

Schmidt and colleagues clarify the role of AIM2 and NLRP3 in sensing dsDNA poxviruses VACV and MPXV in different primary human cell types. The approaches to do this are elegant and use cutting edge technology of employing inhibitory nanobodies to implicate AIM2. The novel aspects of the study are: (1) The generation, characterisation and use of AIM2 nanobodies (2) Clarification of the cell type specific role of AIM2 and NLRP3 in inflammasome sensing of VACV (3) Demonstration of inflammasomes involved in sensing monkey pox virus (4) a role for the poxvirus immune evasion protein poxin in inhibiting NLRP3 inflammasome activation (presumably by inhibiting upstream cGAS function).

We would like to thank the reviewer for the overall positive assessment of our work.

There are a few simple experiments that would help further clarify their model:

1. Show in each cell type where IFN γ renders cells able to sense VACV via AIM2 that AIM2 protein expression is actually increased by IFN γ . Currently this is only shown for THP-1 cells. Given part of the novelty of the study is the clarifying work in primary human cells, AIM2 protein expression should be shown in these (MDMs, NHEKs)

To complement the qPCR analysis, we now also demonstrate the upregulation of AIM2 by IFN- γ using immunoblot with anti-AIM2 antibodies (see new Fig. 6, C and D). While the increased AIM2 protein levels were sufficient for detection in lysates of THP-1 cells and primary Normal Human Epidermal Keratinocytes (NHEKs) (Fig. 6C), we immunoprecipitated AIM2 using AIM2 nanobodies to reveal the upregulation in primary monocytes and macrophages (Fig. 6D).

2. B2R encodes poxin which degrades cGAMP, but many viral immune evasion proteins have multiple functions, so the relationship between B2 effect and a role for cGAS may not always be direct. Given the importance in the paper of clarifying the role of AIM2 versus cGAS-STING-NLRP3 for inflammasome activation, it would be good to compare the B2R mutant virus to WT virus not only in the cell types where NLRP3 is implicated a central, but also in the cell types where AIM2 and not NLRP3 is implicated as central.

During the revision experiments, we have re-sequenced the originally generated recombinant VACV strain VACV Δ B2R C1C-EGFP and found that the initial isolate was unfortunately not a clean knockout for B2R. Instead of replacing the B2R gene with a selection marker as intended, the entire plasmid for homologous repair was in this case inserted into the viral genome (this option is typically eliminated by linearizing the template plasmid). As the plasmid still contains the selection marker surrounded by the B2R flanking regions, this was not apparent in our first Sanger sequencing reactions but only became evident when we sequenced the broader genomic locus. We have now constructed two independent, clean B2R knock strains that are validated by complete sequencing of the surrounding genomic region (and the absence of the negative selection GyrB-PKR marker encoded in the plasmid backbone). The new VACV WR Δ B2R C1C-EGFP strains still activate AIM2 inflammasomes in THP-1 cells (see Fig. R1A below), but unfortunately the true B2R knockouts do not activate inflammasomes in primary human monocytes, although infection and early gene expression are not affected (see Fig. R1B below). We currently have no explanation for this finding, although it is potentially interesting.

Unfortunately, the lack of inflammasome activation does not allow us to draw any conclusion with regards to the role of B2R in cGAS-STING-dependent NLRP3 inflammasome activation in monocytes. We have therefore decided to remove the VACV Δ B2R data from manuscript (but are happy to describe the findings if the reviewers disagree).

Fig. R1. VACV WR Δ B2R C1C-EGFP triggers AIM2 inflammasomes in THP-1 cells, but no inflammasomes activation in primary human monocytes. (A) PMA-differentiated WT THP-1 cells were either left untreated or pre-treated overnight with IFN- γ and infected with the indicated viruses at an MOI of 5 in the presence of VX and where indicated, CRID3. 6 h post-infection, cells were harvested, fixed, and infection and C1C-EGFP speck assembly was analyzed by flow cytometry. (B) CD14⁺ monocytes were infected with VACV C1C-EGFP or two newly prepared VACV Δ B2R C1C-EGFP strains at an MOI of 10, in the presence of 40 μ M VX. Where indicated, cells were co-treated with 2.5 μ M CRID3, 2 μ M H-151, or 12 μ M G140. Data represent values (with individual data points from N=3 independent donors (CD14⁺ monocytes) \pm SEM, or from N=3 independent experiments \pm SEM (THP-1 cells)).

3. Related to (3), there are many small molecule cGAS and STING inhibitors that can be easily used in the primary cells to further clarify a role for cGAS/STING in poxvirus-mediated inflammasome activation. It would be interesting to see if there is a direct correlation between the role of cGAS/STING and NLRP3 by using cGAS/STING small molecule inhibitors as well as CRID3.

We have performed additional experiments in primary monocytes and found that neither the STING inhibitor H-151, nor the cGAS inhibitor G140 substantially decrease the fraction of speckling cells (see new Fig. 7F), or the uptake of DNA dyes (see new Fig. 7G). This data does not back up the proposed model of cGAS-STING-dependent NLRP3 activation that had been proposed based on data in BlaER-1 cells. We amended the discussion accordingly.

Referee #2:

The manuscript titled "Incoming poxvirus dsDNA genomes differentially activate AIM2 or NLRP3 inflammasomes in human cells" by Tesfamariam et al. explores the molecular pathways leading to inflammasome activation in human primary cells infected with poxviruses. Using innovative tools such as antagonistic AIM2 nanobodies and engineered vaccinia viruses, the authors demonstrate cell-type-specific inflammasome activation: AIM2 inflammasomes are triggered in IFN- γ -treated macrophages and keratinocytes, while NLRP3 inflammasomes are activated in CD14⁺ monocytes. Importantly, the study establishes that incoming poxvirus DNA genomes directly activate AIM2 inflammasomes and are critical for NLRP3 activation in monocytes. These findings provide valuable insights into the mechanisms of poxviral modulation of host immune responses.

Strengths:

The manuscript is well-structured, with clearly presented results supported by a comprehensive experimental approach. It employs diverse methods, including engineered vaccinia viruses, AIM2 nanobodies, and reconstituted cell systems, to dissect the interplay between poxvirus infection and inflammasome activation. The development and characterization of novel AIM2 nanobodies represent a significant contribution to addressing the lack of specific AIM2 inhibitors.

Tesfamariam et al. present a thorough and impactful study on poxvirus-induced inflammasome activation, with significant implications for understanding host-pathogen interactions. Addressing the outlined suggestions would further enhance the manuscript's clarity and scientific impact.

We would like to thank the reviewer for the overall positive assessment of our work.

1. Figure 1. VACV infection induces inflammasome activation in IFN- γ treated macrophage-like cell line. (A-D) The authors used PMA-differentiated THP-1 macrophages, either pre-treated with LPS, IFN- γ , IFN- α , followed by infection with VACV WR WT. They measured IL-1 β in the supernatant, LDH release, membrane permeabilization quantified by the uptake of DNA dye. The results showed that in IFN- γ pretreated macrophage-like cells, VACV WR WT infection induced inflammasome activation leading to IL-1 β secretion and cell death via pyroptosis. (E) The authors generated a THP-1 reporter cell line expressing caspase-1CARD-EGFP (C1C-EGFP), which allowed quantification of inflammasome assembly upon LPS plus Nigericin treatment. I am not sure why the authors did not include an example of dot plot of flow analysis of IFN- γ -pretreated THP-1 reporter cells infected by VACV WR WT in Figure 1E. But they did include a quantification of % speckling cells in Fig. 1F.

We agree with the reviewer's suggestion and now use VACV-infected THP-1 C1C-EGFP cells as an example to visualize the distinct population of specking cells due to the redistribution of C1C-EGFP fluorescence (see new Fig. 1E).

2. (G) showed engineering of VACV expressing C1C-EGFP or EGFP. They showed that VACV C1C-EGFP infection of THP-1 WT cells pre-treated with IFN- γ induced IL-1 β and LDH release. Is there any difference between VACV WR WT vs. VACV C1C-EGFP in inducing IL-1 β secretion, LDH release and uptake of DNA dye in IFN- γ -pretreated THP-1 cells?

We have now conducted additional experiments in which THP-1 cells were in parallel infected with VACV WT and VACV C1C-EGFP (Fig. EV1). The expression of C1C-EGFP from an early promoter did not affect IL-1 β release and cell death as assessed by LDH release.

3. Figure 2. Poxvirus infection induces inflammasome activation in primary human cell types. The authors investigated whether vaccinia infection in primary human cell types, including human monocytes-derived macrophages (differentiated in GM-CSF or M-CSF), normal human epidermal keratinocytes (NHEKs), CD14⁺ monocytes induce inflammasome activation in comparison to PMS-differentiated THP-1 macrophage-like cells, with or without IFN- γ pre-treatment. The labels for the condition did not appear until Fig. 2E. It would be helpful to include the labels in Fig. 2A or 2B.

We agree with the reviewer and have re-organized the labels to enhance clarity.

4. The authors did not provide a mechanistic explanation why GM-CSF and M-CSF hMDMs are different in their response to VACV C1C-EGFP in inducing inflammasome activation.

In the initial set of inflammasome experiments with monocyte-derived macrophages, we did not observe any inflammasome assembly in M-CSF macrophages and did not include them in further experiments (although they were responsive to other inflammasome triggers that were assessed in different projects). In later experiments conducted during the revision, we found that IFN- γ -treated M-CSF macrophages from different donors were able to assemble inflammasome upon VACV infection and this was not sensitive to the NLRP3 inhibitor CRID3. Likely, M-CSF macrophages activate AIM2 as described for GM-CSF macrophages and keratinocytes. We do not know whether this is donor-dependent or if there are technical differences (we e.g. worked with different batches of FBS and M-CSF in between both series of experiments), but we think that the different responses between distinct donors and macrophages subtypes are beyond the scope of this study. To avoid absolute statements on M-CSF macrophages that may mislead the field, we have removed the initial panel on primary human M-CSF macrophages. This does not change the interpretation of any of our findings in the manuscript

Fig. R2 Vaccinia virus infection induces inflammasome activation in primary M-CSF differentiated macrophages. Human monocytes-derived macrophages (hMDMs) differentiated in M-CSF were left untreated or pre-treated with IFN- γ overnight. Cells were infected with VACV C1C-EGFP at an MOI of 5 in the presence of VX, and, where indicated, 2.5 μ M CRID3. Cells were harvested and fixed 6 h post infection and EGFP⁺ cells were analyzed for infection and C1C-EGFP speck assembly by flow cytometry. Donor 1-3 = batch 1 and donor 4-6 batch = 2.

- In addition, unlike in Figure 1, the authors did not mention about IL-1 β secretion, LDH release and uptake of DNA dye in this experiment with human primary cells. They only focused on the formation of specks in the infected cells

We invested a substantial amount of time to develop the inflammasome reporter C1C-EGFP to quantify inflammasome assembly at single cell resolution. The intention was to validate the assay in a well-controlled system and then focus on a single, robust, and quantitative method to probe inflammasome assembly once it was established in principle. The reporter is meanwhile also described in a preprint (<https://www.biorxiv.org/content/10.1101/2025.03.30.646205v1>). In our hands, cell death (in particular in response to virus infection) is less specific for inflammasome assembly, while the detection of pro-inflammatory cytokines such as IL-1 β not only relies on inflammasome assembly, but also on the transcriptional upregulation of pro-IL-1 β . However, we do agree that it is informative to perform additional assays in primary cells to understand the physiological response to virus-mediated inflammasome assembly. We measured DNA uptake as a readout for GSDMD pore formation (and/or cell death) (see new Fig. 2, B, E, and H as well as Fig. EV2, A, C, and E) and found that it matched inflammasome assembly. We also quantified LDH release (see Fig. EV2 H, J, and L), but found no substantial increase, likely as this bulk assay is not sensitive enough to detect a small fraction of dying cells. We lastly also determined IL-1 β levels by HTRF but could not detect any IL-1 β in VACV-infected primary cells (see Fig. EV2G, I, and K). VACV can interfere with innate immune sensing on multiple levels, for example by expressing broad-specific caspase inhibitors, soluble IL-1 receptors, and many factors that interfere with the induction of type I and II interferons and the expression of NF- κ B-driven genes. We assume that the transcriptional upregulation of pro-IL-1 β is blocked by VACV infection. In this case, as pointed out by reviewer #3, inflammasome assembly could only be studied by detecting ASC specks with C1C-EGFP, as this readout is as upstream as possible and therefore less sensitive to host-modulatory factors.

- In Fig. 2F and 2G, the authors compared the infection efficiency and inflammasome induction of MPXV and VACV in NHEK cells and THP-1 cells expressing C1C-EGFP. They found that MPXV infection is weaker in inducing inflammasome activation compared with VACV. It would be helpful to evaluate IL-1 β secretion, LDH release and uptake of DNA dye in this experiment to get a full picture of inflammasome activation.

Experiments with MPXV must be conducted in a certified BSL3 laboratory, which limits the number of available methods. We were not able to conduct experiments that required the transfer of supernatants and subsequent washing steps in an open system in our BSL3 laboratory. We also do not have access to an Incucyte Live-Cell Analysis System for cell death analyses in our BSL3 facility. In this context, measuring an early step of inflammasome assembly in a cell-based assay is beneficial when working with BSL3 pathogens like MPXV, as fixed cells can be safely exported from the BSL3 laboratory. As discussed in the revised manuscript, detection of specks is also more robust in case pathogens interfere with aspects downstream of inflammasome assembly.

7. In Fig. 2H, the authors examined induction of specks by VACV C1C-EGFP and VACV EGFP in THP-1 cells with various deficiencies in innate immune sensors with or without IFN- γ pre-treatment. They found that in THP-1 cells pre-treated with IFN- γ , the induction of specks is dependent on AIM2 and ASC, but not on NLRP3. Again, it would be helpful to include IL-1 β secretion, LDH release and uptake of DNA dye in this experiment.

Our previous experiments in THP-1 cells have established that VACV infection induces genuine inflammasome assembly with multiple readouts, and that quantification of C1C-EGFP specks was indicative inflammasome assembly. We therefore felt that it is not necessary to measure additional readouts downstream of inflammasome assembly to determine the involved sensor upstream of ASC speck formation in the same cell line.

8. Figure 3. Identification and characterization of AIM2 nanobodies. This is the novel aspect of this study. The authors provided rationale for developing AIM2 nanobodies for this study, as there are no available AIM2-specific inhibitors. They immunized alpaca with MBP-AIM2PYD L10AL111A, a mutant of the PYD that cannot polymerize into filaments, and cloned sequences for the variable domains of heavy chain-only antibodies (VHHs) into a phagemid vector. They used phage display method to identify AIM2PYD-specific nanobodies. A total of 18 hits were identified. Out of the 18 candidates, three nanobodies were selected, VHHAIM2-1, VHHAIM2-2, VHHAIM2-3, based on their function in AIM2 binding, but not NLRP1CARD binding. Competitive ELISA was performed using VHH-LPETG, which revealed that VHHAIM2-1 and VHH AIM2-3 share an overlapping epitope. Does the author know the identity of the epitope recognized by both VHHAIM2-1 and VHH AIM2-3?

As part of a collaboration with the Hao Wu lab, we have also solved the crystal structures of the three nanobodies in complex with the AIM2^{PYD}, which will be published in a separate manuscript (spearheaded by the involved postdoctoral scientist in the Wu lab). In line with the biochemical data, VHH_{AIM2-2} binds to an epitope of the AIM2^{PYD} that is distinct and non-overlapping with that of VHH_{AIM2-1} and VHH_{AIM2-3}. The latter two bind to the PYD in an almost identical manner, which matches with their sequence similarity. The identified epitopes also fit with the observed inhibition of AIM2^{PYD} oligomerization. Yet, we do not think the exact structure is relevant for any of the conclusions we draw in this manuscript. We provide a figure describing the structure of the nanobody:AIM2^{PYD} complexes as a 'related manuscript' to avoid showing unpublished data in a rebuttal letter that will be shared with the public in case the manuscript is accepted.

9. Figure 4. AIM2 nanobodies inhibit inflammasome assembly by preventing oligomerization of AIM2PYD filaments. The authors used a transfection system to evaluate whether cytosolic expression of AIM2PYD nanobodies interfered with AIM2 inflammasome assembly and AIM2PYD-EGFP filament formation. The results confirmed that AIM2 nanobodies, VHH_{AIM2-1}, VHH_{AIM2-2}, VHH_{AIM2-3}, either monovalent or bivalent, are able to inhibit AIM2 inflammasome activation and AIM2PYD filament oligomerization. Would it be possible to use a biochemical method to show AIM2PYD filament formation and disruption in the presence of AIM2 nanobodies, in addition to the imaging?

To address inhibition of oligomerization with an independent biochemical assay, we exploited the fact that recombinant MBP-AIM2^{PYD} is soluble, but oligomerizes into (insoluble) AIM2^{PYD} filaments once the MBP fusion partner is cleaved off with Tobacco Etch Virus (TEV) protease. The filaments can be separated from soluble monomers and small oligomers by high-speed centrifugation, followed by analysis of pellets and supernatants by SDS-PAGE and Coomassie staining. In the presence of bivalent control nanobody, the entire pool of AIM2^{PYD} was sedimented after removal of MBP, i.e. all AIM2^{PYD} monomers oligomerized into insoluble filaments found in the pellet (see new Fig. 4G and H, Fig. EV4A). In the presence of the bivalent inhibitory nanobodies VHH_{AIM2-1}-VHH_{AIM2-1} and VHH_{AIM2-2}-VHH_{AIM2-2}, AIM2^{PYD} complexed with nanobodies stayed in suspension (i.e. in the supernatant) and almost no AIM2^{PYD} was found in the pellet. We also conducted the same experiments with monovalent nanobodies, which showed the same trends. Yet, the experiments could not be interpreted as clearly, as monovalent nanobodies and AIM2^{PYD} have nearly the same molecular weight and cannot be distinguished by SDS-PAGE (our AIM2 antibodies exhibited substantial background when used to detect bacterially expressed AIM2^{PYD}, precluding analysis of AIM2^{PYD} levels by immunoblot in this set of experiments).

The manuscript describing the VHH structures in complex with AIM2^{PYD} will also include electron microscopy experiments which directly shows that nanobody binding prevents the formation of AIM2^{PYD} filaments (see figure in the 'related manuscript').

10. Figure 5. VACV encoded antagonistic AIM2 nanobodies reveal cell-type-specific inflammasome activation in human primary cells. The authors engineered a vaccinia virus (VACV) strain that expresses C1C-EGFP under the J2R early promoter and bivalent nanobodies driven by a synthetic early/late promoter. Their findings demonstrate that VACV expressing a bivalent control VHH targeting NP1 failed to inhibit speck formation. In contrast, VACV expressing bivalent VHHS targeting ASC or AIM2-1 effectively blocked speck formation in GM-CSF-derived human monocyte-derived macrophages (GM-CSF-hMDMs) and normal human epidermal keratinocytes (NHEKs). However, in CD14⁺ monocytes, VACV expressing bivalent ASC inhibited speck formation, whereas VACV expressing bivalent AIM2-1 did not. Notably, speck formation in CD14⁺ monocytes was effectively inhibited by the NLRP3 inhibitor CRID3. These results suggest that VACV infection triggers distinct inflammasome activation pathways depending on the cell type. In IFN- γ -treated GM-CSF-hMDMs and NHEKs, VACV induces AIM2-dependent inflammasome activation, whereas in IFN- γ -treated CD14⁺ monocytes, inflammasome activation occurs via NLRP3. It would be helpful to present representative images in addition to the quantification graphs.

We thank the reviewer for the suggestion and have now added representative images of infected primary macrophages, NHEKs, and monocytes with C1C-EGFP specks (see new Fig. 2, C, F, and I, as well as figure EV2). Please note that the experiments were conducted in the presence of VX-765 to block caspase-1 activity and pyroptotic cell death, which allows us to stain intact cells containing inflammasomes.

11. Figure 6. VACV infection is directly sensed by reconstituted AIM2 inflammasomes. The authors demonstrated that IFN- γ treatment induces AIM2 expression in various primary cells and PMA-differentiated THP-1 cells. While AIM2 RNA levels were shown for primary cells, both RNA and protein levels were presented for THP-1 cells. It would be helpful for the authors to also include protein level data for AIM2 in CD14⁺ monocytes, NHEKs, GM-CSF-macrophages, and M-CSF macrophages, with and without IFN- γ treatment, to provide a more comprehensive analysis.

To complement the qPCR analysis, we now also demonstrate the upregulation of AIM2 by IFN- γ using immunoblot with anti-AIM2 antibodies (new Fig. 6, C and D). While the increased AIM2 protein levels after IFN- γ treatment could be clearly detected in lysates of THP-1 cells and primary Normal Human Epidermal Keratinocytes (NHEKs) (Fig. 6C), we had to immunoprecipitate AIM2 using our customized AIM2 nanobodies to reveal the upregulation in primary monocytes and macrophages (Fig. 6D).

12. To further investigate AIM2 function, the authors generated THP-1 AIM2 knockout (KO) cells and transduced them with doxycycline (dox)-inducible AIM2 constructs. They observed that VACV-C1C-EGFP infection in Dox-treated THP-1 cells induced speck formation irrespective of IFN- γ treatment. This finding suggests that overexpression of AIM2 can bypass the requirement for IFN- γ priming.

Additionally, the authors reconstituted Dox-inducible AIM2 and NLRP3 in HEK293 cells engineered to express ASC-EGFP but lacking cGAS and STING. Upon infection with VACV or MPXV, they found that both viruses induced speck formation in HEK-AIM2i cells, but not in HEK-NLRP3i cells, in the presence of Dox. These results indicate that viral DNA directly activates AIM2 inflammasomes in these reconstituted cells. It would be helpful to present representative images in addition to the quantification graphs.

We have provided representative images of primary cells expressing the C1C-EGFP reporter as this potentially provides additional insights due to the morphology of the (infected) primary cells. We believe that not every experiment needs to be repeated during the revision process, particularly if the additional readouts are unlikely to yield new or meaningful insights. HEK293T reporter cells reconstituting inflammasomes incorporating fluorescent fusions of ASC are regularly reported in the literature (e.g. PMIDs: 24919149, 36315050, 36941400). We therefore concluded that they do not need further verification.

13. Figure 7. VACV incoming genomes are sufficient for the activation of inflammasomes, independent of viral DNA replication.

Lastly, the authors evaluated whether incoming viral genomes are sufficient to activate inflammasomes. To test this, they used the translation inhibitor cycloheximide (CHX) to block all viral gene expression and the DNA replication inhibitor cytosine arabinoside (AraC) to inhibit viral DNA replication and intermediate and late gene expression. In THP-1 cells constitutively expressing C1C-TagBFP and pre-treated with IFN- γ , infection with VACV expressing EGFP under an early promoter induced inflammasome activation. This activation was blocked by CHX but not by AraC, indicating that incoming viral genomes are sufficient to activate the AIM2 inflammasome in differentiated THP-1 cells. Furthermore, confocal imaging revealed that released viral DNA co-localized with C1C-mCherry. It would be helpful to include quantification data from this experiment and also include imaging from CHX-treated samples.

We have now conducted additional microscopy experiments and have quantified the number of C1C-mCherry specks that occurred in close proximity ($< 1 \mu\text{m}$) to a released viral genome (see new Fig. 7D). We found that around 20% of all specks were next to released viral genomes in both untreated and AraC-treated cells. As discussed in more detail in response to comment 15, CHX-treated cells do not permit early viral gene expression and will not express the inflammasome reporter C1C-mCherry. It will therefore not be informative to include images of CHX-treated cells, as they will only show viral cores that cannot undergo uncoating (as well documented in the literature, see e.g. PMID 23891003). To test whether the co-localization of inflammasomes and released genomes may also occur by chance, we infected AIM2 knockout cells with VACV C1C-mCherry (to provide the reporter but avoid inflammasome assembly) and initiated NLRC4 inflammasome assembly by delivering *Shigella flexneri* MxiH into the cytosol using the anthrax toxin delivery system. Hardly any NLRC4 inflammasomes were found in close proximity to released viral genomes, supporting that the observed co-localization was indicative of genuine binding of viral DNA to AIM2.

14. The authors then investigated which viral structures trigger inflammasome assembly in CD14⁺ monocytes, given that activation occurs through the NLRP3 inflammasome in these cells. They observed that VACV Δ B2R C1C-EGFP induced a higher percentage of specks compared with VACV C1C-EGFP, and this effect was moderately reduced in the presence of the STING inhibitor H-151. AraC treatment did not affect speck induction by either VACV C1C-EGFP or VACV Δ B2R C1C-EGFP. To provide a comprehensive understanding of inflammasome activation by VACV C1C-EGFP and VACV Δ B2R C1C-EGFP in CD14⁺ monocytes, it would be important to include measurements of IL-1 β secretion, LDH release, and uptake of DNA dye in this experiment.

We have conducted all the requested assays to test cell death and IL-1 β release in VACV-infected primary cells in figures 2 and EV2. Please refer to our response to reviewer 1 with regards to the data with VACV Δ B2R. Treatment with the STING and cGAS inhibitors H-151 and G140, respectively, did not substantially reduce specking responses and uptake of DNA dyes (see Fig. 7, F and G). This is now discussed in the revised manuscript.

15. Additionally, incorporating CHX treatment in CD14⁺ monocytes would help demonstrate the importance of viral gene expression in NLRP3 inflammasome activation in these cells.

We had included infection experiments of monocytes in the presence of CHX. However, as cycloheximide also inhibits expression of the reporter C1C-EGFP, we could not quantify inflammasome assembly as we did in the other conditions. It is well-established that release of viral genomes requires viral gene expression (PMID: 14126296, 23084750). As Gaidt et al. (PMID: 29033128) had shown that transfection of dsDNA is sufficient to activate NLRP3 inflammasomes in primary monocytes (which could be quantified by IL-1 β secretion in the absence of viral host-modulatory factors) and as DNA replication was not required, we think that the data is adequate to draw the conclusion that incoming genomes are required and sufficient for inflammasome assembly in monocytes as well.

Referee #3:

In this manuscript, the authors dissect inflammasome activation in various cell types as an immune response to Vaccinia virus and Mpox virus. Within immortalized and primary myeloid cell types, AIM2 inflammasomes are activated. However, in monocytes, VACV activates the NLRP3 inflammasome. The investigators develop novel AIM2 nanobodies, designed to block the polymerization and nucleation of inflammasomes - it stands as the first specific inhibitor of AIM2. The authors utilize novel techniques such as their Casp-1 CARD EGFP construct inside of the VACV genome to assess inflammasome activation - and was necessary because of additional immunomodulators within the VACV/Mpox genome can alter the readout of other assays such as pyroptosis and IL-1B release. The incorporation of the AIM2 nanobody sequence into the VACV genome was also an interesting experimental design - which worked to show the AIM2 dependence in some cell types. Lastly, the activation of inflammasomes was shown to be a result of initial infection, virus-replication independent - suggesting the AIM2 binds directly to the viral genome. The NLRP3 inflammasome activation is modulated by the activity of host STING, where VACV-encoded poxins-schlafen proteins interfere with STING cross-talk with NLRP3 inflammasomes and lysosomal rupture. All-in-all, this investigation yields critical insights into the immunomodulation of host cells by VACV, and provides a mechanistic explanation for inflammasome activation in various cell types.

- specific major concerns essential to be addressed to support the conclusions
No major concerns were noted.

- minor concerns that should be addressed

1. Three separate instances throughout the paper suggest that experiments were performed and data was collected but not shown. There is also a manuscript in process that is cited early on regarding the development of in-house nanobodies? I would suggest to compile this data into an additional supplementary figure.

The mentioned manuscript on the C1C reporter is meanwhile on BioRxiv (<https://www.biorxiv.org/content/10.1101/2025.03.30.646205v1.full>) and is cited in the manuscript. There is no separate manuscript on other in-house nanobodies that are relevant for this study.

Regarding the absence of inflammasome assembly in VACV-infected BLaER1 cells: We had initially infected BLaER1 cells with VACV C1C-mCherry and observed robust infection, but no inflammasome assembly (see Figure R3 below for the reference of the reviewers). While we did multiple experiments, we did not have adequate repeats that would be required to publish the data in a supplementary figure. Meanwhile, it became clear that BLaER1 cells are contaminated with Squirrel Monkey Retrovirus (https://edoc.ub.uni-muenchen.de/29067/1/Xiao_Qianhao.pdf). We do not think that retrovirus-infected BLaER1 cells are a good model to obtain clean answers for innate immune responses. We did not want to take the risk of transferring this retrovirus to other cell lines just to have additional repeats of this negative data. We would therefore prefer to not show the data in the supplementary figures but indicated the data as preliminary in the revised text.

Fig. R3: BLaER1 monocytes do not mount an inflammasome response to VACV infection. WT BLaER1 cells were left untreated or treated with IFN- γ overnight, and infected with VACV C1C-mCherry for 6 h at an MOI of 5 in the presence of VX, and where indicated, CRID3. Cells were fixed in 4% PFA, and infection (A) and inflammasome assembly (B) were quantified by flow cytometry. Data shown are from one representative experiment ($N = 1$).

We had constructed recombinant VACV strains expressing both the inflammasome reporter C1C-EGFP and the Galectin3-mCherry reporter for disrupted endo-/lysosomes. As a positive control, we infected human monocytes and treated cells with Leu-Leu-OMe (LLME) to induce lysosomal rupture (see figure R4, B). Indeed, we did observe Galectin-3-mCherry positive structures (but no equivalent structures in cells just expressing mCherry), indicating that the reporter works as intended. However, when cells were infected and analyzed in the absence of drugs, or after direct activation of STING using diABZI, we did not observe clear Galectin-3-mCherry structures. Nevertheless, we felt it would require quantitative image analysis to make absolute statements on the absence of disrupted endosomes or lysosomes after VACV infection and/or DNA exposure. As the monocytes were technically quite difficult to image because they are not adherent and required special chambers from ibidi for imaging, we felt that more extensive experiments and analysis were not justified solely to support the negative data.

Figure R4: VACV infection does not induce lysosomal membrane permeabilization in primary CD14⁺ monocytes. Primary CD14⁺ monocytes were infected with VACV E C1C-EGFP EL Galectin-3-mCherry (MOI 10) for 6 h in the presence of VX. Where indicated, cells were at the same time stimulated with LLME for 1 h (A, B), with diABZI (A, C) for 6 h, or left untreated (A, D). EGFP expression and speck formation was analyzed by flow cytometry (A). Cells were fixed, stained for DNA, and representative confocal microscopy images were recorded (B-D). Please note the appearance of Galectin-3-mCherry puncta in virus-infected, LLME-treated cells (panel B, top). No comparable structures were observed when a control virus just expressing mCherry was treated with LLME (panel B, bottom). We did not observe comparable structures either when cells were only infected (D), or infected and treated with the STING agonist diABZI (C). Average values (with individual data points) from N=3 independent donors (A) ± SEM are displayed. Scale bar = 5 μm.

‘No co-localization of nanobodies or AIM2 with poxvirus genomes was observed in conditions where nanobodies prevented AIM2 inflammasome assembly.’

This sentence refers to pilot experiments that we conducted to test if we can visualize endogenous AIM2 accumulating on incoming viral DNA in the presence of inhibitory nanobodies (which would avoid the formation of ASC specks, which tend to non-specifically bind to many antibodies or nanobodies, hampering clean analysis of microscopy data). The initial rationale was based on the old model that AIM2^{HIND200} binds to dsDNA once inhibition by AIM2^{PYD} was relieved. We did not observe any accumulation of endogenous ASC on dsDNA revealed with fluorescent nanobodies against the AIM2^{PYD}. This could be interpreted as evidence for a model in which AIM2^{HIND200} binding to dsDNA requires formation of AIM2^{PYD} filaments (which cannot occur in the presence of the inhibitory nanobodies). However, there are of course also alternative explanation for the absence of staining. As we did not extensively

test different conditions to stain endogenous AIM2 with the indicated nanobodies, we decided to omit this sentence from the manuscript.

Dear Florian,

Thank you again for the submission of your revised manuscript (EMBOJ-2024-119572R) to The EMBO Journal for our consideration, and for your patience during peer review. Your manuscript has been sent back to two of the three original referees who had previously assessed the first version of the work (referees #1 and #2), and we have now received their comments, which are included below.

I am very pleased to say that both referees are satisfied with the revision and recommend publication without further comments. In light of this input, I am glad to inform you that your manuscript has been accepted in principle for publication in our journal - congratulations on an excellent work!

Regarding your response to the previous comments of referee #3, I would like to thank you for your detailed and justified answer. In line with the referee's comment, however, I must inform you that as per our journal's policy, "data not shown" or "preliminary data" are not permitted. We kindly request that you provide all mentioned data in the Appendix (alternatively, please provide the citation of the source where they can be found, if they are already published elsewhere), or otherwise remove all mentions of data that are not important for the study's conclusions.

There are also a few more changes we need from you to make in the final version of your manuscript, before we can proceed with its formal acceptance and publication:

- Please move "Funding agencies" to the "Acknowledgements" section of your revised manuscript.
- The "Summary" above the Abstract must be removed. Instead, this text could be used for the short synopsis summary of the findings and significance (please see below for more information).
- Please provide a list of up to 5 relevant keywords (preferably broad terms to enhance online search discoverability of your article) after the Abstract.
- Heading "Materials and Methods" should be renamed to "Methods".
- Thank you for providing links for reviewers to access your entries in the Nanosaurus database. You have noted that these links are not for use in publication, therefore we kindly request you to update this section including the permanent IDs and specific URLs to all datasets, which must be publicly available at the time of publication.
- We also noticed that access information to your mass spectrometry data is missing from the Data availability statement. Please include in the revised statement the database, IDs, and specific URLs to these data (which must also be publicly available at the time of publication).
- Heading "Declaration of interests" should be renamed to "Disclosure and competing interests statement".
- We noticed that you have uploaded Figure EV2 in two parts, namely "Figure EV2_A-N" and "Figure EV2_O-P". Please make sure to upload the Figure as a single file instead. If this is not possible, you might consider moving some panels to the Appendix (but in that case, all Figure callouts throughout the manuscript should be updated accordingly).
- The title page of the Appendix PDF file should contain the heading "Appendix for:", followed by the manuscript's title and a brief Table of Contents including the page numbers of all listed items; the nomenclature should be "Appendix Figure S#" or "Appendix Table S#" throughout the manuscript and the Appendix PDF file.
- Please note that EMBO press papers are accompanied online by:
 - A) a short (2 sentences) summary of the findings and their significance,
 - B) 2-5 short bullet points highlighting the key results, and
 - C) a synopsis image in .jpg or .png format that is exactly 550 pixels wide and 300-600 pixels high (the height is variable). Please note that all text needs to be legible at the final size.Please upload this information along with your revised manuscript (the text for A and B should be provided in a separate Word file).
- During our standard Figure checks, our team detected image reuse between Figure 2C, F & I AND Figure EV2 B, D & F. We note that this reuse is not listed in the Figure legends. Please check these Figures carefully, revise them if necessary, or -if the reuse is intentional and justified- please clarify and mention/explain the reuse in the Figure legends.
- Please revise the headings and order of the manuscript's sections as follows: Title page - Abstract - Keywords - Introduction - Results - Discussion - Methods - Data Availability - Acknowledgements - Disclosure and Competing Interests Statement -

References - Figure Legends - main Tables (if applicable) - Expanded View Figure Legends.

Please also note that as part of the EMBO publications' Transparent Editorial Process, The EMBO Journal publishes online a Peer Review File along with each accepted manuscript. This File will be published in conjunction with your paper and will include the referee reports, your point-by-point response and all pertinent correspondence relating to the manuscript. You can opt out of this by letting the editorial office know (contact@embojournal.org). If you do opt out, the Peer Review File link will point to the following statement: "No Peer Review File is available with this article, as the authors have chosen not to make the review process public in this case."

We look forward to seeing a final version of your manuscript as soon as possible. Please let us know if you have any questions and use this link to submit your revision: <https://emboj.msubmit.net/cgi-bin/main.plex>.

Best regards,

Ioannis

Referee #1:

MY original comments stand on the novelty of the paper. The authors have addressed the issues raised by reviewers in as much as is possible.

Referee #2:

The authors have addressed all of my comments adequately.

All editorial and formatting issues were resolved by the authors.

Dear Florian,

Congratulations on an excellent manuscript! I am very pleased to inform you that it has been accepted for publication in The EMBO Journal. Please excuse the delay in this final decision, and accept our thanks for comprehensively addressing the initially raised referee concerns and our editorial requests for changes and corrections.

If you have any questions, please do not hesitate to contact the Editorial Office. Thank you for your contribution to The EMBO Journal. Working with you has been a pleasure.

Best regards,

Ioannis

Please note that it is The EMBO Journal policy for the transcript of the editorial process (containing referee reports and your response letters) to be published as an online supplement to each paper. If you should prefer removal of any referee-only figures included in the point-by-point response(s), e.g. because they may still be used for future publication or because they have been reproduced from published work by others, please do let us know immediately via response email. More information is available here: https://www.embopress.org/transparent-process#Review_Process
